# Refractory inflammatory arthritis definition and model generated through patient and multi-disciplinary professional modified Delphi process

**Hema Chaplin**[1]*, **Ailsa Bosworth**[2], **Carol Simpson**[3], **Kate Wilkins**[3], **Jessica Meehan**[1], **Elena Nikiphorou**[3], **Rona Moss-Morris**[1], **Heidi Lempp**[3‡], **Sam Norton**[1,3‡]

**1** Health Psychology Section, Institute of Psychiatry, Psychology and Neuroscience, King's College London, London, United Kingdom, **2** National Rheumatoid Arthritis Society, White Waltham, United Kingdom, **3** Centre for Rheumatic Diseases, King's College London, London, United Kingdom

‡ HL and SN are joint last authors on this work.
* hema.chaplin@kcl.ac.uk

**Data Availability Statement:** All relevant data (raw and summary) are within the paper and its Supporting Information files.

## Abstract

### Objective

Various definitions have been proposed for Refractory Disease in people with Rheumatoid Arthritis; however, none were generated for Polyarticular Juvenile Idiopathic Arthritis or involving adult and paediatric multidisciplinary healthcare professionals and patients. The study aim is to redefine Refractory Disease, using Delphi methodology.

### Methods

Three rounds of surveys (one nominal group and two online (2019–2020)) to achieve consensus using a predetermined cut-off were conducted voting on: a) name, b) treatment and inflammation, c) symptoms and impact domains, and d) rating of individual components within domains. Theoretical application of the definition was conducted through a scoping exercise.

### Results

Votes were collected across three rounds from Patients, Researchers and nine multi-disciplinary healthcare professional groups (n = 106). Refractory Inflammatory Arthritis was the most popular name. Regarding treatment and inflammation, these were voted to be kept broad rather than specifying numbers/cut-offs. From 10 domains identified to capture symptoms and disease impact, six domains reached consensus for inclusion: 1) Disease Activity, 2) Joint Involvement, 3) Pain, 4) Fatigue, 5) Functioning and Quality of Life, and 6) Disease-Modifying Anti-Rheumatic Drug Experiences. Within these domains, 18 components, from an initial pool (n = 73), were identified as related and important to capture multi-faceted presentation of Refractory Inflammatory Arthritis, specifically in Rheumatoid Arthritis and Polyarticular Juvenile Idiopathic Arthritis. Feasibility of the revised definition was established

**Funding:** This paper represents independent research funded by the National Institute for Health Research (NIHR) Maudsley Biomedical Research Centre at South London and Maudsley NHS Foundation Trust and King's College London, in the form of a PhD Studentship (IS-BRC-1215-20018) for the first author (HC). The funders had no role in study design, data collection and analysis, decision to publish, or preparation of the manuscript.

**Competing interests:** The authors have declared that no competing interests exist.

(2022–2023) with good utility as was applied to 82% of datasets (n = 61) incorporating 20 outcome measures, with two further measures added to increase its utility and coverage of Pain and Fatigue.

## Conclusion

Refractory Inflammatory Arthritis has been found to be broader than not achieving low disease activity, with wider biopsychosocial components and factors incorporating Persistent Inflammation or Symptoms identified as important. This definition needs further refinement to assess utility as a classification tool to identify patients with unmet needs.

## Introduction

Refractory Disease (RD) in people with rheumatoid arthritis (RA) refers to the persistence of disease activity and symptoms despite treatment with multiple drugs with different mechanisms of action [1], by not achieving low disease activity target after resistance to two or more biologics [2–5]. Previous definitions of RD have focused on biological underpinnings based mainly on rheumatologists' experiences alone (if definition generation stated) [1], without consideration of wider psychosocial contextual components that need to be incorporated from a multi-disciplinary perspective [6]. Therapeutic strategies following a treat-to-target regimen are generally successful in reducing inflammatory markers, however patient-reported outcomes have not similarly improved [7, 8].

There is evidence that patients' prefer holistic approaches adopting patient-centred care [9] to avoid the misattribution that symptomatology is solely due to inflammation. This absence of adequate management strategies ensures persistent poor patient-reported outcomes. This current project, along with others [2, 5], stems from a Versus Arthritis workshop in 2015 on the topic of RD that highlighted the need for a broader definition, incorporating the patient perspective [10]. A key knowledge gap identified was differences in patients and clinicians' definitions of RD. The biopsychosocial model of RD is poorly understood and refining the concept could identify novel approaches to adopt a broader perspective to treatment and care.

Although current RD research focuses on Rheumatoid Arthritis (RA), there is clear justification for incorporating those diagnosed with Polyarticular Juvenile Idiopathic Arthritis (PolyJIA) who are now in adulthood, who despite presenting a RD course are commonly excluded from such research therefore outcomes and impact of disease and treatment are not fully understood and under recognised [1, 11]. Since JIA, and in particular the Polyarticular subset, are not benign and self-limiting to childhood [12], it is important and clinically useful to utilise a common language and approach to classify and treat these patients similarly to other rheumatic diseases seen in adult rheumatology care [13].

Absence of a systematic approach to identify or evaluate RD means the true impact and underlying mechanisms remain largely unknown [5]. The growing number of publications on this topic [1] and European Alliance of Associations for Rheumatology (EULAR) Task Force on the wider concept of Difficult-to-Treat RA [2], highlights the need to identify, consolidate and implement additional components of RD. This could assist identifying patients who may benefit from pharmacological versus non-pharmacological interventions, and address additional strategies, such as coping and self-management.

The aim was to explore and refine the definition of Refractory Disease (RD) considering the perspectives of healthcare professionals (HCPs) and patients, explore its meaning and

implications, and identify uncertainties about the terminology and components. The primary objective was to reach a revised consensus definition of RD in RA and PolyJIA from across multi-disciplinary HCPs and patients' perspectives. The secondary objective was to assess the feasibility of using the revised definition to identify RIA in a theoretical application, and revise associated outcome measures if necessary.

## Materials and methods

### Design

The current study was designed and set up following the Versus Arthritis workshop [10], comprising of a PhD studentship initiated in 2017. The study involved three phases: 1) Component Development and 2) Delphi Voting (Fig 1). The Strengthening the Reporting of Observational Studies in Epidemiology (STROBE) statement guidelines [14] have been adhered to, where appropriate, and checklist for cohort studies included. Full NHS ethical approval granted in July 2018 by London–Hampstead Research Ethics Committee (18/LO/1171).

In phase 1, a pool of components was developed based on a multimodal evidence synthesis. We use the term components to refer to specific facets that operationalise RD (e.g. synovitis, joint pain).

In phase 2, a consensus definition of RD was developed using a Delphi approach, by surveying panels of experts to achieve a group agreement or convergence of opinion [15]. Components not included in the consensus definition were retained in a broader conceptual model.

In phase 3, a theoretical application of the definition was conducted through a scoping exercise of rheumatology registries and cohorts.

### Component development

Components of RD were identified from: 1) qualitative interviews and focus groups with patients and HCPs [16], 2) systematic review of studies of RD in RA/PolyJIA [1], and 3) review and application of biopsychosocial theories regarding chronic illness and persistent symptoms. Firstly, an initial inductive thematic analysis of 25 patient (RA and Adult PolyJIA) and 32 HCP interviews conducted between August 2018 and April 2019 identified 17 components for common experiences of RD and persistent symptoms, whilst an additional seven were patient-specific (including JIA-specific) and four professional-specific [16] (see unpublished thematic map in S1 Fig). This qualitative data analysis was part of a larger framework analysis, which

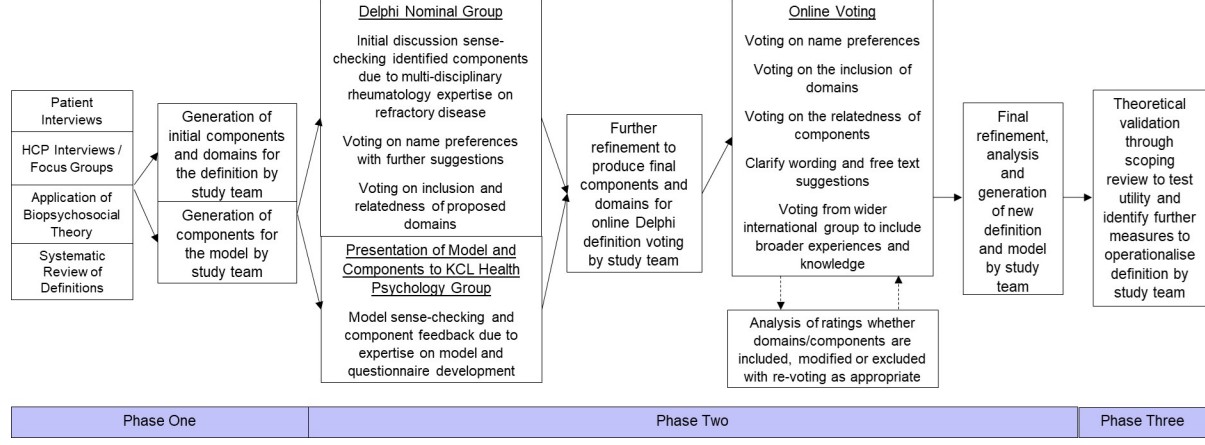

**Fig 1. Study overview.**

involves five steps [17, 18]: (i) familiarisation with data, (ii) preliminary thematic analysis to develop initial themes (presented here and [16]), with further steps occurring later which were: (iii) application of themes to the whole dataset systematically, (iv) reducing data from transcripts into summaries and organising these into a matrix (participants by themes), and (v) identifying patterns and relationships across participants and themes (manuscript in preparation). Secondly, a systematic review of existing definitions [1] identified three key elements of RD: 1) treatment, 2) presence or absence of inflammation and 3) symptoms and impact. These elements were used to group the components and a three-part definition was explored. Due to the large number of components identified, and to account for similarities between components, these were grouped thematically into higher level domains (e.g. disease activity, pain).

From the interviews and systematic review, it became clear that some factors (e.g. reduced mobility) are related to defining RD, whilst others (e.g. social support) are more relevant to describing poor prognostic, perpetuating or protective elements related to the experience of RD. These factors can guide treatment strategies in line with a biopsychosocial formulation [19–21]. This is an integrative process that enables consideration of the complex, interacting factors implicated in development of a patient's presenting problems to translate the diagnosis into specific, individualised interventions. This formulation has not previously been applied to RD and could help holistically define and explain RD drawing from theoretical models such as Adjustment to Chronic Illness and Illness Perception Models [22, 23] by aligning identified concepts to established theories and utilising their terminology and descriptions. These theoretical models outline the individual behavioural, emotional and cognitive factors involved in adjustment and adaptation to chronic illness related to the return to equilibrium following critical illness events/stressors.

Findings from these three areas of research were combined during phase 1 to develop an initial 72 components across 12 domains (see S1 Data) for the Delphi phase [24], by mapping these concepts onto currently used measures within Rheumatology [25] such as Musculoskeletal Health Questionnaire (MSK-HQ) [26] and (Child) Health Assessment Questionnaire ((C) HAQ) [27, 28] to increase utility and validity. The specific wording of the components was agreed by the wider study team (HC, SN, HL, CS, KW and EN).

## Delphi voting

The Delphi method allows synthesis of the best available information, an anonymous and iterative process of consensus and validation between key stakeholders/end-users, with feedback thereby increasing ownership and engagement [29]. Here we used a modified Delphi process through an initial face-to-face nominal/expert group (September 2019), followed by two rounds of Delphi surveys online (March-May 2020) [15, 24, 30–32] to vote on preferred terminology and variables that define RD. Specifically, we sought consensus on (i) preferred name, (ii) domains to retain in the definition, and (iii) components to retain within each domain. The process was managed by a non-voting co-ordinator (HC).

**Participants and sample size.** A purposive sampling strategy [33] was employed to identify potential Delphi participants who had experience of RD and/or persistent symptoms in RA/JIA. Three independent inclusion criteria determined who was invited [33]: (1) Patients with RA/PolyJIA (determined by diagnosis, regardless of age) or HCPs within Rheumatology, (2) involvement with RA/JIA patient organisations or (3) recognized academic career in RD or Persistent symptoms in RA/JIA. Participants were contacted via e-mail and invited to participate anonymously online via Qualtrics software (Qualtrics 2020, Provo, UT), to take part in all online rounds of voting, regardless of whether they had participated in the previous round.

This was to ensure a wide representation and allow recruitment to target during COVID-19 since many academics and HCPs were redeployed to frontline NHS services during this time (March and May 2020). Written informed consent was obtained prior to participation in each round.

There is currently no formal sample size calculation in Delphi processes [15]. A pragmatic approach was adopted in line with previous literature [30–32]. The absolute minimum sample size agreed was 12 per round, with at least three members per participant group (Rheumatologists, other HCPs, Patients, and Researchers). A target sample size of at least 40 was set for the online surveys based on estimated 95% confidence intervals (CIs) providing sufficient precision to discriminate between domains rated for inclusion/exclusion from the consensus definition. A total of 89 participants were invited to allow for non-response and attrition.

**Data collection and analysis.** Four different stages of analysis were conducted: 1) Name ranking, 2) Treatment/Inflammation cut-off inclusion, 3) Symptoms/ Impact domain inclusion, and 4) Component importance rating. Firstly, proportions and 95% CIs were calculated to determine the name preference voting. Participants indicated their top three name choices in each round with the choice receiving majority vote in the final round proposed as the final definition name. Secondly, for the treatment and inflammation questions, there was a need to achieve consensus on whether the definition needs to specify distinct cut-offs or include a broader statement (see S2 Data). Options receiving most responses in the first round were taken forward to the next round. Thirdly, for each domain participants were asked to rate whether the domain needed inclusion on a nine point Likert scale (1 = 'Definitely Not Include' to 9 = 'Definitely Include') [15]. In line with previous literature [30–32], a priori level of agreement was required ($\geq$70% rating domains 7–9 for inclusion and 1–3 for exclusion). Mean and 95% CIs were also estimated to allow consideration of the certainty of domain inclusion or exclusion.

Finally, for each component, participants were asked to rate how related the component is to assess a) Refractory Arthritis, and b) Disease flare (-3 = 'Highly Unrelated' to 3 = 'Highly Related'), and to provide comments to clarify component meaning [30] (see S2 Data). Inclusion of components in the consensus definition was based on: a) high mean related total scores for Refractory Arthritis, and b) mean difference and effect sizes between ratings for Refractory Arthritis and Disease Flare. This second criterion allowed discriminant validity to be established to ensure selected components appropriately differentiated between these two constructs. Scores within participant groups were also considered to identify differences across groups and ensure that components rated as important by one group, particularly patients and rheumatologists, were not excluded.

Several exploratory subgroup analyses were undertaken to assess differences across the four different role groups (Patients, Rheumatologists, Other HCPs and Researchers), in terms of representation across the different rounds of voting in both those who started and completed all voting, and in ranking and scoring of the symptoms and impact domains for inclusion. To assess whether there were any differences across the different rounds of voting regarding the sociodemographic characteristics such as years of rheumatology experience and adult versus paediatric trained, one-way ANOVAs were conducted for continuous variables and Fishers exact tests for categorical variables due to low numbers per category. To assess differences across the clinical role groups (Rheumatologists, Patients and Other Healthcare Professionals) in voting on domains for inclusion, simple linear regressions for each domain were conducted with additional tests and contrasts to determine significant variability.

The minimum response for each domain was set at $\geq$75% [32] and only whole domains could be missing rather than components within domains due to the software used for data collection, further analysis such as mixed-effect models was not required to adjust for missing

data. Additional analysis of the related ratings to generate Item-level Content Validity Indexes (I-CVI) [34] was conducted to ensure the final definition is composed of an appropriate components to adequately represent the construct of interest whilst also considering inter-rater agreement to reflect consensus. The final chosen components were then again initially mapped onto to routinely used measures identified from literature searching [35, 36].

### Theoretical definition validation

A theoretical validation through literature scoping of worldwide RA and JIA registries/cohorts was undertaken (between October 2022 and February 2023) to highlight what components and domains of the definition could be utilised, and to identify alternative measures if possible. The feasibility and application of using the identified outcome measures or data points to identify RIA was then explored using frequencies. Websites of relevant rheumatology organisations were screened for details of relevant registries and/or cohorts with PROMs for RA and JIA. This was supplemented by internet searches of rheumatology registries and cohorts to identify details of the data collected in the registries which was extracted from study-specific websites or publications. This initial searching uncovered summary studies that have a similar scoping aim and captured some of the required data [36–39]. A summary table was produced to highlight the data captured across the included registries and cohort studies [36].

### Patient and public involvement

Patients were involved in the design, conduct, reporting, and dissemination of this research, in which AB, CS and KW have been integral at all stages as Patient Research Partners. Numerous others were involved in the design of this project which evolved through various stages of involvement with people with RA and PolyJIA. Both diagnoses were included because early patient and public involvement work revealed that RD is a problem across both age and disease groups.

## Results

### Sample characteristics

Across the three rounds,106 votes were cast in this Delphi study as seen in Table 1. Due to anonymity, the total number of unique participants or number of rounds each participant voted in cannot be determined. There were no differences in participant characteristics (e.g. Rheumatology experience or diversity of roles) between each round for those that completed, nor between those who dropped out (n = 3 each online round) (see S2 Fig). In Round One, duplicate data for two responses (e.g. started survey on one device then continued on another) was removed, otherwise attrition occurred such that domain completion rates were between 90–100%, therefore meeting the pre-determined minimum response rate. In Round Two, the attrition occurred after the name ranking questions, with otherwise complete data.

### Name preference voting

During the nominal group workshop (one missing voter due to late arrival), ten terms were voted on for the name of the definition, of which RD, Persistent Disease and Treatment Resistant were the most popular (see S3 Data). Moreover, participants suggested that the terminology should include Part A and Part B elements to form different name iterations to be voted on subsequently, with the following additions: 'Ongoing', 'Inflammatory Arthritis', 'Syndrome', 'Inflammation' and 'Symptoms'. Consequently, the top three preferences from Round One online voting were combined to create nine names involving all combinations of the label

**Table 1. Characteristics of Delphi voters (n = 106).**

| | Face-to-Face Nominal Group (N = 13) | Online Delphi Round One (N = 40) | Online Delphi Round Two (N = 53) |
|---|---|---|---|
| Role, n (%)† | | | |
| Rheumatologist | 5 (38.5%) | 11 (27.5%) | 17 (32.1%) |
| Nurse | 1 (7.7%) | 4 (10%) | 3 (5.7%) |
| GP | 1 (7.7%) | 3 (7.5%) | 3 (5.7%) |
| Pharmacist | 0 | 1 (2.5%) | 3 (5.7%) |
| Psychologist | 0 | 2 (5%) | 3 (5.7%) |
| Podiatrist | 0 | 1 (2.5%) | 1 (1.9%) |
| Physiotherapist | 0 | 3 (7.5%) | 5 (9.4%) |
| Occupational Therapist | 0 | 2 (5%) | 4 (7.6%) |
| Patient Representatives | 3 (23.1%) | 8 (20%) | 10 (18.9%) |
| Researcher* | 3 (23.1%) | 4 (10%) | 3 (5.7%) |
| Social Worker | 0 | 1 (2.5%) | 1 (1.9%) |
| Number of years' experience in Rheumatology, mean (SD) | | | |
| Professionals | 15.10 (12.07) | 15.59 (11.30) | 16.07 (8.62) |
| Patients | 21.33 (16.29) | 25.62 (14.17) | 28 (12.29) |
| Predominantly trained/worked‡, n (%) | | | |
| Adult | 9 (90%) | 20 (62.5%) | 29 (67.4%) |
| Paediatric | 1 (10%) | 6 (18.75%) | 7 (16.3%) |
| Both | 0 | 6 (18.75%) | 7 (16.3%) |
| Received specific MSK training‡, n (%) | | | |
| Yes | 6 (60%) | 24 (75%) | 32 (76.7%) |
| No | 4 (40%) | 8 (25%) | 10 (23.3%) |

†Although participants were allowed to select multiple roles, the main role is reported here.

‡Professional roles only were reported.

*Researchers covered the following disciplines: Radiology, Psychology, Medical Sociology, Rheumatology and Health Services, Statistics and Epidemiology.

(e.g. refractory, resistant) and target (e.g. inflammatory arthritis, disease), which participants were asked to again select their top three in Round Two (see S3 Data). Refractory Inflammatory Arthritis received the majority vote (25% of votes), followed by Persistent Inflammatory Arthritis (19% of votes). There was slight disparity in the top preference between Patients and the other three groups as Persistent Inflammatory Arthritis received 21.6% of the votes by Patients whilst both RIA and RD received 15%.

## Definition of refractory inflammatory arthritis

Based on the Delphi process (detailed below), a consensus definition for RIA is proposed in Table 2. This three-part definition covers the core elements identified in phase 1: 1) treatment, 2) inflammation presence (Persistent Inflammation) or absence (Persistent Symptoms) and 3) symptoms and impact, which comprises 18 components across 6 domains to capture the presentation and experience of RIA. To identify a patient as having RIA, at least one component from each part 1–3 need to be met, and initial suggestions for assessment for each component included 20 data points (see S5 Fig) such as (C)HAQ [27, 28], MSK-HQ [26], Consultation / Joint Examination / Clinical Notes [40] and EuroQol 5-Dimensions (EQ5D) [41]. These are based on commonly used measures in paediatric and adult rheumatology clinical cohorts and

**Table 2. Revised consensus definition of refractory inflammatory arthritis.**

| REFRACTORY INFLAMMATORY ARTHRITIS | |
|---|---|
| Part 1: Treatment | DESPITE following Treat-to-Target Strategy using ≥1 csDMARDs, and ≥1 anti-TNF, bDMARDs and/or tsDMARDs with different mechanisms of action |
| Part 2: Inflammation | Synovial inflammation is PRESENT (Persistent Inflammation) or ABSENT (Persistent Symptoms) as determined by inflammatory markers, physical examination and imaging or composite disease activity |
| Part 3: Symptoms and Impact | PLUS ≥1 of the following: |
| | 1. Disease Activity<br>a. Persistently high inflammation and/or symptoms e.g. with or without fluctuations for at least two consecutive clinical visits over the period of at least six months<br>b. Disease Activity not captured by DAS28 (hands, shoulders, wrists, elbows and knees) including involvement of other joints (hips, temporomandibular, feet), extra-articular manifestations or other inflammatory features (vasculitis, uveitis, tendonitis or enthesitis) or non-inflammatory features (muscle weakness or cachexia)<br>c. Repeated need of short course steroid tablets or intra-articular injections due to inflammatory arthritis manifestations, that may or may not control flare and localised swelling |
| | 2. Joint Involvement<br>a. Joint stiffness during the day (longer than 30-60 minutes upon wakening)<br>b. One or two persistently active/affected joints despite acceptable control in other joints<br>c. Accrued damage due to inflammation - Joint erosion(s), deformity(ies) or restrictions in range of movement |
| | 3. Pain<br>a. Pain in joints e.g. hands and feet<br>b. Pain during the day and/or night<br>c. Pain Interference impacting on quality of life |
| | 4. Fatigue<br>a. Lack of physical energy resulting in difficulties conducting daily activities e.g. washing, dressing<br>b. Lack of mental energy leading to difficulties with concentration and memory |
| | 5. Functioning and Quality of Life<br>a. Problems with self-care e.g. washing/dressing<br>b. Inability to perform desired activities e.g. hobbies, social, work<br>c. Poor physical function e.g. lack of strength, dexterity, grip<br>d. Reduced mobility and/or Problems walking, standing, or climbing stairs e.g. driving, use of public transport, needing to sit<br>e.Disease-related distress e.g. psychological distress related to burden of disease including Physical related, Emotional, Social, Treatment or Healthcare Distress |
| | 6. cs/b/tsDMARD Experiences<br>a. Primary inefficacy (no response to DMARD at all) and/or Secondary inefficacy (developed DMARD resistance over time)<br>b. Experience of multiple occurrences of inefficacy, intolerability, or discontinuation |

registers [35, 36], with suggested cut-offs or scoring direction on continuous measures and were explored further in the theoretical validation exercise.

An overview of the phases of definition development including refinements and voting results during Delphi rounds are presented in S3 Fig. Nominal group attendees agreed all 12 domains (85–100%) to be included in the next stage of voting. All domains were voted as related to RD, with Pain, DMARD Experiences, Joint Activity and Co-morbidities rated the most highly, whilst Poor Quality Sleep, Steroid Use and Dependency and Social Functioning and Quality of Life were rated the lowest. Discussion included the addition of Healthcare Utilisation (eight components) to the Steroids domain, with refinements/additions to eight other components e.g. sex, benefits, variability, assessment of feet and removal of one component ("unable to relax easily"). They also stated the importance of investigating other undiagnosed underlying reasons for these symptoms in case these treatable causes are driving suspected RD

e.g. thyroid issues causing fatigue. Further discussions with the study team and Health Psychology Group at KCL took the final total count of components identified for voting to be 73 across 10 domains as the three Functioning and Quality of Life Domains were condensed into one domain (see S1 Data).

In Round One online voting, for Parts One and Two regarding Treatment and Inflammation factors, the consensus was for these to be kept broad (65–70%) as originally proposed rather than specifying fixed cut-offs (Tables 2 and 3). As seen in Fig 2, seven domains in Round One met consensus criteria for inclusion (≥70% rating the domain between 7–9 for inclusion on the 1–9 scale) for Part 3 Symptoms and Impact (see S1 Table for data and decisions). Of these, three were clearly above this inclusion threshold, whilst the remaining four indicate uncertainty as to the generalisability of inference. The remaining three domains did not reach agreement whether to include or exclude, and when converted into rankings were consistently the lowest across the different role groups (see S4 Fig). Rheumatologists and Other Healthcare Professionals ranked Joint Activity highest, then Stiffness and Disease Activity in their top three. Whereas Patients ranked Disease Activity and Fatigue jointly highest with Pain third as their top three, therefore more symptom focussed. Researchers ranked Disease Activity highest then Pain and Functioning and Quality of Life as joint second in their top three.

Regression analyses revealed no differences in mean include scores across the training specialties (Adult vs Paediatric vs Both), however there were some subtleties in the domains reaching criteria for inclusion. Analyses highlighted the different priorities and considerations for HCPs with paediatric versus adult patients. For Paediatric professionals, Fatigue (66.67%) and DMARD experiences (33.33%) did not reach inclusion criteria, with Pain (66.67%) and Functioning and Quality of Life (66.67%) also falling short for those trained in Both. There was greater variability in responses for the group that had received both Adult and Paediatric training, who were heterogenous population that encompassed a mixture of Researchers, Rheumatologists, a Physiotherapist and a GP.

From Round One voting, 30 components from the seven included domains were retained based on mean related scores and the difference with flare, and I-CVI scores. The core study team (HC, SN, HL) then discussed and prioritised those with higher mean related score, followed by those with medium-to-large difference whilst considering the I-CVI and any free text comments regarding modification and clarification (see S4 Data & S5 Fig). This resulted in final data driven selection of 16 components, through combining two domains (Joint Activity and Stiffness into Joint Involvement) and combining some components (e.g. Pain during the day and Pain during the night combined into Pain during day and/or night).

Thirteen components, not achieving consensus in round one, were taken forward to Round Two (see S1 Data): nine components from Sleep, Co-morbidities and Wider Involvement Outside of Joints and Healthcare/Medication Utilisation and exploration of four exclusion criteria from components voted in Round One as highly unrelated. From Round Two, only two components out of the 13 voted upon reached the required criteria for inclusion (see S1 Table and Fig 2), taking the total number of components achieving consensus for inclusion to 18. The core research team decided not to hold a third round of online voting as only three components had not achieved consensus for inclusion or exclusion. Given each included domain consisted of at least two components, these three components were excluded from the definition.

## Refractory Inflammatory Arthritis (RIA) conceptual model

A wider conceptual model was generated by the core research team to allow for consideration of the relationship between the domains and components excluded from the consensus

**Table 3. Assessments mapped onto refractory inflammatory arthritis definition.**

| RIA Definition | | Assessment Methods (including cut-offs) |
|---|---|---|
| Part 1: Treatment | | Treatment History (≥1 csDMARDs, and ≥1 anti-TNF/b/tsDMARDs) [40] |
| Part 2: Inflammation | | Abnormal/Raised Inflammatory markers (ESR / CRP based on the local laboratory standards or ESR >10 and CRP >3 based on mean values) [40, 107]<br>Physical examination (T/S/A/LJC) ≥1 [40, 108]<br>Imaging of Synovial Inflammation (Ultrasound or MRI) [40] |
| Part 3: Symptoms and Impact | 1. Disease Activity | Clinical notes / Consultation (or mention of extra-articular manifestations etc) / Medical and Treatment History [40] DAS28 >3.2 [109] or SDAI >11 [110]<br>RAID (scoring higher on Q5) [111] |
| | 2. Joint Involvement | MSK-HQ (scoring 0–2 on Q1 and/or 2) [26]<br>Consultation / Physical examination (TJC/SJC) ≥1 / Clinical Notes [40, 108]<br>Stiffness Assessment (>30 minutes duration) [112]<br>Imaging of Damage/Erosions Score (Radiographs) [40] |
| | 3. Pain | MSK-HQ (scoring 0–2 on Q1 and/or 2) [26]<br>Pain VAS >3.5 [113] or PGA ≥3 [106]<br>EQ5D (scoring 2 or 3 on Q4) [41]<br>RAID (scoring higher on Q1) [111]<br>Pain Scale of SF-36 (average of recoded items 21 and 22) [105] |
| RIA Definition | | Assessment Methods (including cut-offs) |
| Part 3: Symptoms and Impact | 4. Fatigue | BRAF (Higher total score and on items) [114]<br>Fatigue VAS >2.0 [115] or PGA ≥3 [106]<br>MSK-HQ (scoring 0–2 on Q10) [26]<br>RAID (scoring higher on Q3) [111]<br>Vitality Scale of SF-36 (average of recoded items 23, 27, 29 and 31) [105] |
| | 5. Functioning and Quality of Life | (C)HAQ (≥1 total score to represent moderate-to-severe disability) [116]<br>EQ5D (scoring 2 or 3 on Q1-3 and Q5) [41]<br>Physical Functioning Scale of SF-36 (average of recoded items 3–12) [105]<br>Role Limitations Scale of SF-36 (average of recoded items 13–19) [105]<br>Social Functioning Scale of SF-36 (average of recoded items 20 and 32) [105]<br>WSAS (higher total score and on items 1–4) [117]<br>MSK-HQ (scoring 0–2 on Q3-7 and Q11) [26]<br>RAID (scoring higher on Q2 and Q6) [111]<br>RADS [118]<br>Emotional wellbeing scale of SF-36 (average of recoded items 24–26, 28, 30) [105] or SF-36 (scoring ≤38 on the Mental Component Subscale) [66]<br>Consultation [40] |
| | 6. cs/b/tsDMARD Experiences | Clinical notes / Treatment and DAS28 History / Consultation (or mention of steroid use and reasons for DMARD changes) [40] |

Please Note: DAS28: Disease Activity Score 28 joint count, SDAI: Simplified Disease Activity Index, RAID: Rheumatoid Arthritis Impact of Disease, (C)HAQ: Child or Adult Health Assessment Questionnaire, MSK-HQ: Musculoskeletal Health Questionnaire, VAS: Visual Analogue Scale, EQ5D: EuroQol 5-Dimensions, BRAF: Bristol RA Fatigue Scale, WSAS: Work and Social Adjustment Scale, RADS: Rheumatoid Arthritis Distress Scale

definition and RIA, based on the Phase 1 development work [1, 16]. The concepts and examples given here are based on findings from the systematic review highlighting the role of other contributing factors such as Serology and fixed disease factors such as disease duration [1] and qualitative work on patients' and HCP' experiences of RIA [16] (see S1 Fig) such as social

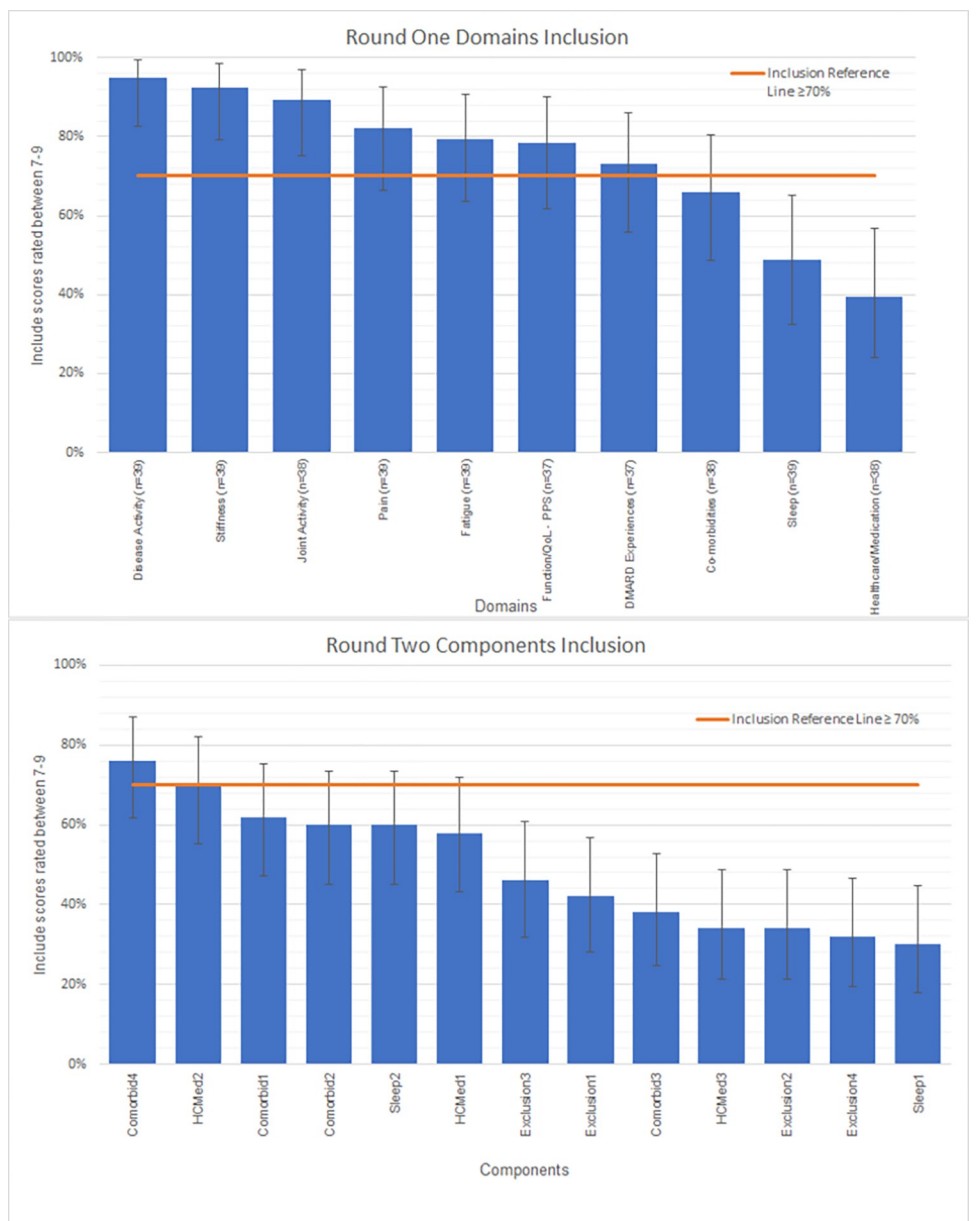

**Fig 2. Round one domains and round two components inclusion.**

support and independence which included elements such as adjustment, resilience and illness/treatment beliefs which align with general theories of perceptions and adjustment to long-term illness. This followed a Biopsychosocial Formulation [19, 20] (see Fig 3), with the Definition of RIA with Persistent Symptoms or Inflammation at the core, surrounded by Predisposing, Poor Prognostic, Perpetuating and Protective factors.

Some of the Precipitating factors are the same to RA/PolyJIA generally, whilst Poor Prognostic factors suggest some mechanisms identified for those that then lead to have a more refractory course, which therefore cannot be modified. However, building on these symptoms and impact are the Protective and Perpetuation factors that describe some of the biopsychosocial mechanisms underpinning why some patients may experience negative and worsening

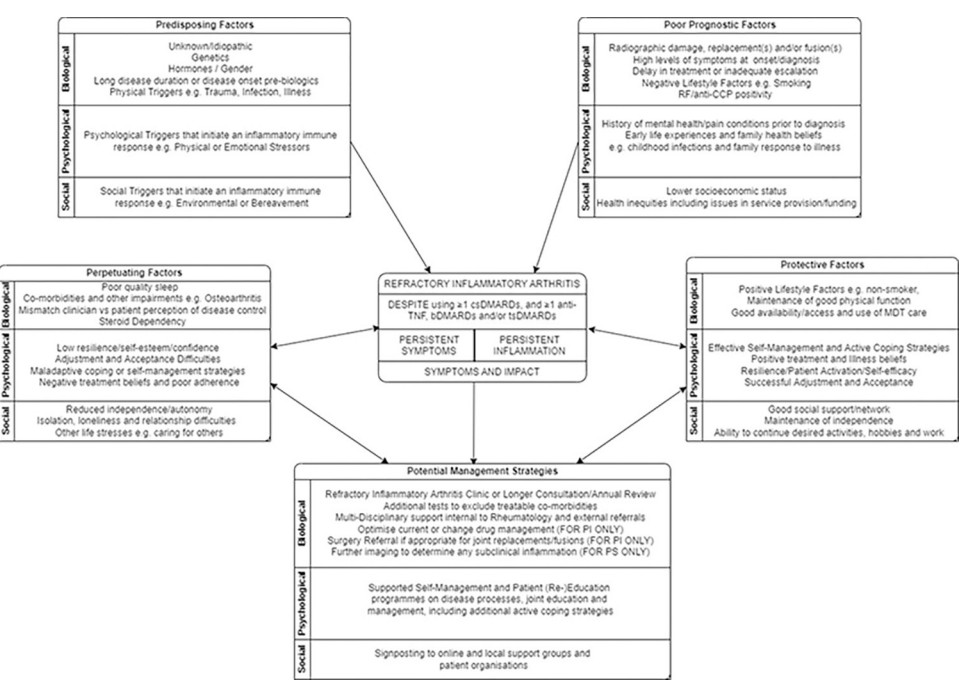

**Fig 3. Proposed conceptual model of refractory inflammatory arthritis.**

impact of RIA (Perpetuating) on their daily lives whilst others appear to manage and live with reduced impact (Protective), despite experiencing RIA with Persistent Symptoms or Inflammation. The allocation of factors is individual and based on the Impact Triad [42], which includes the patient's perceived severity, importance and ability to self-manage/cope with the factor or symptom to determine patient priorities from the level of perceived impact. The Protective and Perpetuating factors can be modified and targeted to reduce disease impact and improve quality of life. This conceptual model allows for the identification of potential risk factors and interventional targets that Rheumatologists and other HCPs can consider as part of the shared decision-making process of deciding which treatments may be most appropriate for patient.

## Theoretical definition validation

A total of 61 registries/cohorts were identified across 63 publications in the scoping exercise [36, 37, 43–104] (detailed in S2 Table), of which most were RA (n = 43, 70.5%) and in Europe (n = 39, 63.9%). Across the registries/cohorts identified, the four most complete components from the 20 initial data points (see S3 Table) were Joint Count (n = 61, 100%), ESR/CRP (n = 60, 98%), (C)HAQ (n = 58, 95%) and Clinical notes/information (n = 55, 90%) with the four most missing components being the Work and Social Adjustment Scale (WSAS), RA Distress Scale (RADS), and Bristol RA Fatigue (BRAF) which were not collected in any, and MSK-HQ was only collected in one [59].

The registries/cohorts contained between 6–15 (27–68%) of the proposed data components (out of 20). For each part of the definition, the following could be applied: Part One: Treatment (82%), Part Two: Inflammation (100%), Part Three: Symptoms and Impact (100%), and for each subdomain (as only one component is necessary): 3.1) Disease Activity (100%), 3.2) Joint Involvement (100%), 3.3) Pain (87%), 3.4) Fatigue (26%), 3.5) Functioning and Quality of Life (98%), and 3.6) DMARD Experiences (92%). Therefore, the RIA definition could be applied to

82% of the identified registries/cohorts. Several appropriate substitutions were used in these studies of which the most common relevant measures that could be substituted were the Patient Global VAS (PGA) (n = 40) and SF-36 (n = 19). By adding these two measures, the following increases in the application of the definition for subdomains of Part Three were found to be: 3.3) Pain (98%) and 3.4) Fatigue (93%), which is the most marked increase.

This scoping exercise established the theoretical feasibility of applying the RIA definition by highlighting which data points are collected or missing. None collected all the initial 20 measures, although the RIA definition as it is initially proposed could be applied to 50 registries/cohorts (82%) if only one item from each part is satisfied and to 10 registries/cohorts (16%) if an item from each subdomain of Part Three is met to identify RIA. With the addition of PGA and SF-36, 45 registries/cohorts (74%) capture one item from each of the six subdomains of Part Three increasing its utility and coverage of all subdomains as discussed below for symptom-based stratification. Therefore following the scoping exercise, SF-36 and PGA was included in the assessments mapped on to the RIA definition as seen in Table 3, as general measures to substitute for missing disease-specific measures [105, 106]. Disease and symptom specific measures, e.g. Pain VAS rather than PGA, need to be prioritized by assessors but to increase application and utility, relevant generic measures may be appropriate alternatives.

## Discussion

This study presents the development of a consensus definition for Refractory Inflammatory Arthritis (RIA) with Persistent Symptoms or Inflammation across multi-disciplinary HCPs and patients, in line with the study aim and objectives. This three-part definition covers core elements of treatment, presence or absence of inflammation, and symptoms and impact, including: 1) Disease Activity, 2) Joint Involvement, 3) Pain, 4) Fatigue, 5) Functioning and Quality of Life, and 6) DMARD Experiences. This captures the multi-faceted presentation and experience of RIA in RA and PolyJIA populations covering biological and psychological symptoms. This work independently validated and supports the growing body of evidence to understand and refine the concept of Refractory Disease (RD), notably the EULAR Difficult-to-Treat Task Force and others [2, 4, 5, 119–121]. It is worth noting that the current study was designed and set up in 2017, as part of a Health Psychology PhD programme of research before publication of the EULAR Difficult-to-Treat Task Force work. These parallel bodies of work have reached similar conclusions that a broader definition needs to include treatment, symptoms (including inflammation) and impact as part of the RIA concept, however these have been more generally stated in the EULAR definition [2, 120].

Here we suggest an extension to this general criterion recently proposed by detailing a three-part definition identified as important to both patients and multi-disciplinary HCPs with 18 specific components within 6 domains that can be operationalised. Additionally, our definition has been derived through a different evidence synthesis approach, e.g. systematic review, qualitative interviews, application of biopsychosocial theory and Delphi consensus voting, which is a different, more methodologically rigorous approach than a survey, scoping literature review and agreement process (consensus not defined) [2]. Another distinguishing feature of this study is participation of a larger number of patients, multi-disciplinary professionals from nine relevant specialties and researchers across both adult and paediatric services to include PolyJIA within this refractory concept, compared to the Difficult-To-Treat Taskforce which consisted of rheumatologists predominantly, two patient partners, one health professional (not defined), one psychologist, one pharmacist and one occupational therapist [2]. Another feature was the selection of the name for the definition, which fits with the common use of the term Refractory in the literature [1], and the distinction with Difficult-To-Treat Disease [120].

For Paediatric professionals, Fatigue and DMARD experiences did not reach inclusion criteria. However, it is important to bear in mind the lower sample sizes for the Paediatric and Both groups compared to the Adult sample, therefore no firm conclusions can be drawn to suggest these domains and their components would need to be removed for a paediatric specific definition. Given there were no significant differences in the domain means, it seems that the proposed definition based on the whole group scores may be appropriate for use in the paediatric PolyJIA population as well, and requires further exploration. This study has shown that it is feasible to include both adult and paediatric professionals and patients. Given the relative rarity of JIA and those specialising in paediatric and adolescent rheumatology, the use of a transdiagnostic approach in future research can be advantageous.

This scoping exercise established the feasibility of applying the revised definition of RIA in line with the second objective to 61 identified datasets, which found that the definition has good utility. The RIA definition could be applied to 82% of datasets and has been revised to include suggestions for 22 data points that are routinely collected in clinical practice and research studies. The main outcome that limited the application of the definition was Treatment History, as only a single item was required for Part One whereas for Parts Two and Three there were multiple alternatives. This lack of reporting for treatment history would also be a limitation for the other definitions that are based on multiple bDMARDs [2, 4]. The findings fit well with a systematic review of outcome measures used in 88 RA registries and long-term observational studies where disease activity (mainly DAS28) and physical functioning (mainly HAQ) was consistently recorded in included studies whilst there was heterogeneity in the patient-centred outcomes for symptom burden and psychosocial ramifications [2, 122]. This type of validation was not conducted with the original definition of Refractory Disease [3], nor to the same degree with the most recent definitions [2].

Unlike previous definitions [1, 2], the development of the new RIA definition presented in this paper included consideration of content and discriminant validity indicators (I-CVI and mean difference with flare) and therefore provides a tool to appropriately identify and measure RIA, alongside a conceptual model of related factors. This RIA definition could serve as a clinical or research checklist with the components aligned to routinely used measures [35, 36], as listed in Table 3 to enable identification of aspects requiring further support e.g. low HAQ indicating issues with functioning and quality of life (Domain 5) could prompt a referral to Occupational and Physiotherapies [8]. The use of cut-offs increase outcome interpretability, making the RIA definition more meaningful and applicable in both in clinical practice and research, by allowing data to be compared and pooled [123]. A criticism of the Difficult-to-treat definition [2] is the subjective character of the third criterion which meant not all elements could be applied to a dataset as it was deemed too subjective to extract from the available data [124]. Messelink and colleagues also highlighted that whether "the management of patients is perceived as problematic" will most often not be routinely noted in health records and stated this lack of clarity and objectivity to be a key challenge in future research using the Difficult-to-treat definition [2].

Within the RIA model presented here, Protective and Perpetuating Factors suggest some of the underlying biopsychosocial mechanisms that may explain why some patients experience negative and worsening impact of RIA in their lives (Perpetuating) whilst others have minimal or less impact despite experiencing RIA (Protective). By identifying the specific Protective and Perpetuating factors deemed to have an impact through biopsychosocial formulation for each patient [19, 20, 42], it may be possible to address these wider clinical and sociodemographic factors that contribute to non-response to treatment, which is important to identify those requiring additional non-pharmacological support from a multi-disciplinary team [6, 8, 21, 121]. This fits with subsequent EULAR work which identified multiple contributing factors, a

high burden of disease and the heterogeneity of Difficult-To-Treat RA [125], suggesting that these factors should be identified in daily practice in order to tailor therapeutic strategies further to the individual patient with the RIA model providing such a framework.

Several limitations and potential bias within the current study should be discussed. It would have been useful to have a patient and clinician included in the core study team discussions during the delphi voting iterations following nominal group discussions however this process was data driven with little subjective decision making made by the researchers involved. Since most domains met pre-defined inclusion criteria in the first round, the process was less iterative than previous Delphi studies [31, 32], without continuous re-voting on the same components and due to anonymity the team was unable to provide individualised feedback to voters. This may be a consequence of online data collection, rather than a face-to-face meeting with further deliberation. However, the completion of an online questionnaire enabled a larger number of participants to take part, specifically during the first COVID-19 wave. There was a predominant bias towards participants from the UK followed by Europe, in particular adult services, and caution is required for applicability of this definition in other countries and for purely paediatric application. This project determined a preliminary definition of RIA, requiring further validation to explore staging within the definition to account for severity and differences between conditions and countries. The refinement could be achieved through assessment of retrospective cohorts and datasets from the UK, followed by different global cohorts.

## Conclusion

Within Rheumatology, parallel bodies of work have reached similar conclusions regarding the broadening of the Refractory or the wider Difficult-to-Treat concept. The authors provide a different angle to define, measure, and conceptualise Refractory Inflammatory Arthritis (RIA), using health psychology theory across PolyJIA and RA with the input of patients, rheumatologists, and multi-disciplinary HCPs. The multi-factorial definition and model proposed can help to identify patients experiencing RIA where inflammation and/or symptoms persist despite treatment and allow for the consideration of wider contextual factors that may not be targeted by pharmacological treatments. Future work could explore the application and validity of the RIA definition in other Rheumatic Musculoskeletal Diseases, both adult and juvenile variations. Appropriate adult and paediatric measures have been proposed to measure RIA [25], which require further validation to assess practicality and feasibility, and could initiate development of classification criteria.

## Supporting information

**S1 Fig. Key components of refractory disease and persistent symptoms identified by patients and rheumatologists during qualitative interviews.**
(PDF)

**S2 Fig. Analysis of participants across Delphi rounds.**
(PDF)

**S3 Fig. Overview of definition development and refinement.**
(PDF)

**S4 Fig. Round one rankings of domains by role group.**
(PDF)

**S5 Fig. Example selection of components for inclusion or exclusion from included domain.**
(PDF)

**S1 Table. Statistics for round one and two components considered for definition.**
(PDF)

**S2 Table. Assessments initially mapped onto refractory inflammatory arthritis definition.**
(PDF)

**S3 Table.** A) Data capture and synthesis of measures used in the RIA definition across RA/JIA Biologic registries/cohort (Location, Condition and Parts One and Two). B) Data capture and synthesis of measures used in the RIA definition across RA/JIA Biologic registries/cohort (Part Three).
(PDF)

**S1 Data.** Lists of Components and Domains presented during a) Nominal Group, b) Round One Online Voting, and c) Round Two Online Voting.
(PDF)

**S2 Data. Example of Delphi voting questions (Round one online).**
(PDF)

**S3 Data. Options for name preference voting with proportions and 95% confidence intervals.**
(PDF)

**S4 Data. Refractory inflammatory arthritis definition components statistics.**
(PDF)

**S1 File.**
(CSV)

**S2 File.**
(CSV)

## Acknowledgments

Thank you to all the patients and healthcare professionals that took part in the Delphi voting and to the Refractory Delphi Nominal Group which comprised of Ailsa Bosworth, Sam Norton, Heidi Lempp, Katie Bechman, Wendy Olsder, Di Skingle, Flora McErlane, Louise Parker, Elena Nikiphorou, Lianne Kearsley-Fleet, Simon Black, Adam Young, and Kevin Davies. Finally, thank to you to Nora Ng, James Galloway, Ian Scott, Debajit Sen, and Rachel Tattersall for support and help recruiting to the qualitative study. For the purposes of open access, the author has applied a Creative Commons Attribution (CC BY) licence to any Accepted Author Manuscript version arising from this submission.

## Author Contributions

**Conceptualization:** Hema Chaplin, Heidi Lempp, Sam Norton.

**Formal analysis:** Hema Chaplin, Jessica Meehan, Heidi Lempp, Sam Norton.

**Funding acquisition:** Hema Chaplin, Heidi Lempp, Sam Norton.

**Investigation:** Hema Chaplin.

**Methodology:** Hema Chaplin, Rona Moss-Morris, Heidi Lempp, Sam Norton.

**Project administration:** Hema Chaplin, Ailsa Bosworth.

**Supervision:** Ailsa Bosworth, Carol Simpson, Kate Wilkins, Elena Nikiphorou, Rona Moss-Morris, Heidi Lempp, Sam Norton.

**Visualization:** Sam Norton.

**Writing – original draft:** Hema Chaplin, Heidi Lempp, Sam Norton.

**Writing – review & editing:** Ailsa Bosworth, Carol Simpson, Kate Wilkins, Jessica Meehan, Elena Nikiphorou, Rona Moss-Morris, Heidi Lempp, Sam Norton.

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
