## [Decision Letter · Decision Letter 0]

21 Jul 2023

PONE-D-23-16246Refractory Inflammatory Arthritis definition and model generated through patient and multi-disciplinary professional modified Delphi processPLOS ONE

Dear Dr. Chaplin,

Thank you for submitting your manuscript to PLOS ONE. After careful consideration, we feel that it has merit but does not fully meet PLOS ONE’s publication criteria as it currently stands. Therefore, we invite you to submit a revised version of the manuscript that addresses the points raised during the review process.

We look forward to receiving your revised manuscript.

Kind regards,

Ryu Watanabe, M.D., Ph.D.

Academic Editor

PLOS ONE

Journal Requirements:

This paper represents independent research funded by the National Institute for Health Research (NIHR) Maudsley Biomedical Research Centre at South London and Maudsley NHS Foundation Trust and King’s College London, in the form of a PhD Studentship (IS-BRC-1215-20018) for the first author (HC). 

Reviewers' comments:

Reviewer's Responses to Questions

**Comments to the Author**

1. Is the manuscript technically sound, and do the data support the conclusions?

Reviewer #1: Yes

Reviewer #2: Yes

2. Has the statistical analysis been performed appropriately and rigorously? 

Reviewer #1: Yes

Reviewer #2: Yes

3. Have the authors made all data underlying the findings in their manuscript fully available?

Reviewer #1: Yes

Reviewer #2: Yes

4. Is the manuscript presented in an intelligible fashion and written in standard English?

Reviewer #1: Yes

Reviewer #2: Yes

5. Review Comments to the Author

Reviewer #1: The presented manuscript describes in details the Delphi process of establishing the wide definitions concerning refractory inflammatory arthritis. As nicely explained in the introduction, there is a need of defining the refractory disease in children and adults alike, a the study addresses this need in line with the highest standards. My only concern and suggestion is regarding the distinction of polyarticular JIA and RA. While there is an increasing number of papers and opinions which aims to unify the terminology for the inflammatory arthritis in adults and children, there are still many dissimilarities which prevents RA and pJIA to be used interchangeably. Therefore, I suggest to provide a more detailed description of how pJIA patients after they turned 16 were termed by the authors of the paper - as having pJIA or RA. Moreover, there are also some concerns regarding the involvement of pediatric rheumatologists in the study, since as shown in table 1, there was only one (of five rheumatologist) in face-to-face group, with the number not significantly increased in round 1 and 2.

Reviewer #2: First of all I need to congratulate the authors to this tremendous and holistic work that they have conducted over the past years. It is a topic of utmost importance, as the authors nicely outline in their introduction. The manuscript is well written in general, but still needs the attention of the reader to follow all steps that have been undertaken to retrieve the presented results. The limitations that came into my mind during reading overlap with those mentioned in the discussion section. My personal opinion does not 100% overlap with the results, but this is not of relevance for the validity and accuracy of the study conduction.

The authors further present a vast compendium of supplementary material that helps the in depth reader to better understand the process and results.

6. PLOS authors have the option to publish the peer review history of their article (what does this mean?). If published, this will include your full peer review and any attached files.

Reviewer #1: **Yes: **Lovro Lamot

Reviewer #2: No

---

## [Author Response · Author response to Decision Letter 0]

24 Jul 2023

Dear Dr Watanabe and Reviewers,

Thank you for comments and acknowledgment that this papers addresses an important issue, and we welcome the decision of minor revisions.

Please find below our comments in reply to the reviewers’ comments and how we have addressed these in the manuscript.

Please find tracked changes in the manuscript, otherwise the previously uploaded documents remain unchanged:

Journal Requirements:

I have now amended the manuscript to meet PLOS ONE’s style requirements.

This paper represents independent research funded by the National Institute for Health Research (NIHR) Maudsley Biomedical Research Centre at South London and Maudsley NHS Foundation Trust and King’s College London, in the form of a PhD Studentship (IS-BRC-1215-20018) for the first author (HC). 

Thank you for highlighting this and providing appropriate wording which I have included in the revised cover letter.

The ethics statement is now only in the Methods Section of the manuscript (Lines 174-175).

I have checked all the references and reference [88] has been updated to reflect the most recent erratum. Otherwise the reference list is complete and correct. 

Review Comments to the Author

Reviewer #1: The presented manuscript describes in details the Delphi process of establishing the wide definitions concerning refractory inflammatory arthritis. As nicely explained in the introduction, there is a need of defining the refractory disease in children and adults alike, a the study addresses this need in line with the highest standards. My only concern and suggestion is regarding the distinction of polyarticular JIA and RA. While there is an increasing number of papers and opinions which aims to unify the terminology for the inflammatory arthritis in adults and children, there are still many dissimilarities which prevents RA and pJIA to be used interchangeably. Therefore, I suggest to provide a more detailed description of how pJIA patients after they turned 16 were termed by the authors of the paper - as having pJIA or RA. Moreover, there are also some concerns regarding the involvement of pediatric rheumatologists in the study, since as shown in table 1, there was only one (of five rheumatologist) in face-to-face group, with the number not significantly increased in round 1 and 2.

We would like to thank Reviewer 1 for their considered review and would like to address their concerns.

Regarding their first concern, I have included the following descriptions as those with PolyJIA were termed as such regardless of age (not RA) in the Introduction and Methods:

Lines 139-140

“Although current RD research focuses on Rheumatoid Arthritis (RA), there is clear justification for incorporating those diagnosed with Polyarticular Juvenile Idiopathic Arthritis (PolyJIA) who are now in adulthood, who despite presenting a RD course are commonly excluded from such research therefore outcomes and impact of disease and treatment are not fully understood and under recognised [1, 2].”

Line 191

“Firstly, an initial inductive thematic analysis of 25 patient (RA and Adult PolyJIA) and 32 HCP interviews conducted between August 2018 and April 2019 identified 17 components for common experiences of RD and persistent symptoms, whilst an additional seven were patient-specific (including JIA-specific) and four professional-specific [3] (see unpublished thematic map in S1).”

Line 243

“Three independent inclusion criteria determined who was invited [4]: (1) Patients with RA/PolyJIA (determined by diagnosis, regardless of age) or HCPs within Rheumatology, (2) involvement with RA/JIA patient organisations or (3) recognized academic career in RD or Persistent symptoms in RA/JIA.”

Regarding their second concern, about the number of paediatric rheumatologists, I have now acknowledged this limitation in the Discussion, with the following statement:

“There was a predominant bias towards participants from the UK followed by Europe, in particular adult services, and caution is required for applicability of this definition in other countries and for purely paediatric application.”

However I would like to also mention that some of paediatric professionals were also categorised in the “Both” group as they indicated they were trained or worked predominantly in both paediatric and adult services which particularly captures those working in adolescent rheumatology. So the total involvement of paediatric rheumatologists was increased in subsequent rounds (doubling the paediatric only numbers), if taking the “Both” category into account.

Reviewer #2: First of all I need to congratulate the authors to this tremendous and holistic work that they have conducted over the past years. It is a topic of utmost importance, as the authors nicely outline in their introduction. The manuscript is well written in general, but still needs the attention of the reader to follow all steps that have been undertaken to retrieve the presented results. The limitations that came into my mind during reading overlap with those mentioned in the discussion section. My personal opinion does not 100% overlap with the results, but this is not of relevance for the validity and accuracy of the study conduction.

The authors further present a vast compendium of supplementary material that helps the in depth reader to better understand the process and results.

We would like to thank Reviewer 2 for their balanced review and supportive comments.

We look forward to these minor revisions being accepted and publication of the manuscript in due course. Once again thank you for your comments and suggestions to improve this piece of work.

---

## [Editor Report · Decision Letter 1]

25 Jul 2023

Refractory Inflammatory Arthritis definition and model generated through patient and multi-disciplinary professional modified Delphi process

PONE-D-23-16246R1

Dear Dr. Chaplin,

We’re pleased to inform you that your manuscript has been judged scientifically suitable for publication and will be formally accepted for publication once it meets all outstanding technical requirements.

Kind regards,

Ryu Watanabe, M.D., Ph.D.

Academic Editor

PLOS ONE

---

## [Editor Report · Acceptance letter]

31 Jul 2023

PONE-D-23-16246R1 

Refractory Inflammatory Arthritis definition and model generated through patient and multi-disciplinary professional modified Delphi process 

Dear Dr. Chaplin:

I'm pleased to inform you that your manuscript has been deemed suitable for publication in PLOS ONE. Congratulations! Your manuscript is now with our production department. 

Kind regards, 

on behalf of

Dr. Ryu Watanabe 

Academic Editor

PLOS ONE